# Combined Flexible Activation Functions for Deep Neural Networks

## Abstract

Activation in deep neural networks is fundamental to achieving non-linear mappings. Traditional studies mainly focus on finding fixed activations for a particular set of learning tasks or model architectures. The research on flexible activation is quite limited in both designing philosophy and application scenarios. In this study, we propose a general combined form of flexible activation functions as well as three principles of choosing flexible activation component. Based on this, we develop two novel flexible activation functions that can be implemented in LSTM cells and auto-encoder layers. Also two new regularisation terms based on assumptions as prior knowledge are proposed. We find that LSTM and fully connected auto-encoder models with proposed flexible activations provides significant improvements on time series forecasting and image compressing tasks, while layer-wise regularization can improve the performance of CNN (LeNet-5) models with PReLu activation in image classification tasks.

## 1 Introduction

Deep learning is probably the most powerful technique in modern artificial intelligence (LeCun et al., 2015a). One reason is its ability in approximating complex functions with a large but limited number of parameters (Cybenko, 1989; Hornik, 1991), while the regular layer structures make it possible to be trained with efficient back propagation algorithms (Goodfellow et al., 2016).

In a deep neural network, the weights and bias take account of linear transformation of the data flow, while the activation functions bring in non-linearity. It is remarked in (Hornik, 1991) that activation functions do not perform equally well if we take minimal redundancy or computational efficiency into account. Thus the selection of activation function for different tasks is an issue with importance. Traditionally, people train the weights of linear transformations between layers while keeping the activation functions fixed, and usually one identical activation function is used for all the neurons on each single layer. For example, rectifier linear units (Relu) are used as the default choice for the activation in hidden units for feed forward units and a large proportion of convolutional neural networks (Nair and Hinton, 2010), while sigmoid and tanh functions are used where output values are bounded, such as in output layers for classification problems and the gate activations in recurrent cells (Gers et al., 1999; Chung et al., 2014).

The drawback of Relu activation is the issue of dead unit when the input is negative, which makes people introduce functions with non-zero values in the negative range, including leaky-Relu and Elu (Maas et al., 2013; Clevert et al., 2015). On the other hand, explosion or vanishing gradients in back propagation are also issues that harm the performance of model largely due to the shape of activations (Bengio et al., 1994; Hochreiter, 1998; Pascanu et al., 2012), while techniques such as clipping and batch normalization can be implemented to alleviate these issues to some extent (Ioffe and Szegedy, 2015; Lin et al., 2017).

With a large enough neural network and sufficient training time, the model can effectively learn the patterns from data with possible high accuracy, however it is not straightforward to confirm that learning process is the most efficient and the results are the most accurate. One possible solution for accelerating model training is to introduce flexible or trainable activation functions (Agostinelli et al., 2014; He et al., 2015; Chung et al., 2016). Even though this requires higher computing and storing cost that is proportional to the number of neurons, the performance of non-linear activation

function can be largely improved, which could be more efficient than increasing the number of basic model parameters or the number of neurons.

There are existing works trying to promote the predictive performance of deep neural networks based on trainable activation functions. As the Leaky Relu function has a hyper-parameter to be optimized, which is the slope of its negative part, parameterized Relu (PRelu) was proposed to make this slope adapt to the data within specific neurons and be learned during the training process (He et al., 2015). Meanwhile, another study proposes the parameterized version of Elu activation, which introduces two parameters to control the shape of exponential curve in the negative region (Li et al., 2018).

It can also be a blending of different commonly used activations, where the trainable parameters are the weights for the combination components (Sütfeld et al., 2018; Manessi and Rozza, 2018). Since different activation functions can have very similar behavior in some specific regions, a more generative way is to consider their Taylor expansions at $0$ point and use a weighted combination of polynomial functions with different orders instead (Chung et al., 2016). For containing those functions that are not centered at $0$, one choice is to train a piece-wise function adaptively (Agostinelli et al., 2014). The similar effect can be achieved by Maxout activation, which is quite helpful in promoting the efficiency of models with dropout (Goodfellow et al., 2013). Beyond that, there are also studies on making the most of the non-linear properties by introducing adaptation mechanism on the Softmax layers (Flennerhag et al., 2018), which achieve the former state-of-the-art results on several natural language processing (NLP) tasks.

The limitation of existing studies can be illustrated as follows. First, most of existing work focus on some specific forms of parameterized activation functions rather than a more general form. Second, there is a lack of study on flexible versions of bounded activations such as sigmoid and tanh. Third, the experiments of existing work are mainly on convolutional networks rather than other types of architectures such as recurrent networks and auto-encoder networks. In this study, we consider the activation function as a combination of a set of functions following the constraints of several principles. Based on these principles, we develop two novel trainable activation functions that can be introduced to LSTM cells and auto-encoder architecture with significant performance improvement. In addition, layer-wise regularization on activation parameters is introduced to reduce the variance caused by activation functions. Correspondingly, we use three experiments to show the goodness of two novel activation functions and the effect of layer-wise regularization on PRelu activation.

## 2 METHODOLOGY

### 2.1 PARAMETERIZED ACTIVATION FUNCTIONS

In this study, we introduce a general form of parameterized activation functions linearly combined from different types of activation functions or a single activation function with multiple parameters. We assume that the parameters in the combined activation functions can be different for each neuron, which can be trained during the main training process of the model parameters with back propagation.

$$o_i(z, \boldsymbol{\alpha}^i, \boldsymbol{\beta}^i) = \sum_{k=1}^{K} \alpha_{ik} f_k(z, \boldsymbol{\beta}_{ik}), \quad \sum_{k=1}^{K} \alpha_{i,k} = 1 \tag{1}$$

where $i$ indexes the neuron, and $z = z_l = \boldsymbol{W_l} X_{l-1} + \boldsymbol{b_l}$ is the input of the activation layer indexed by $l$. This means that, at each neuron $i$, it is possible to have its own set of parameters $\boldsymbol{\alpha}^i = [\alpha_{i1}, ..., \alpha_{iK}]^T$ and $\boldsymbol{\beta}^i = [\boldsymbol{\beta}_{i1}, ..., \boldsymbol{\beta}_{iK}]$ where $\alpha_{ik}$ is the combination weights and $\boldsymbol{\beta}_{ik}$ is the activation parameter vector for the $k$-th component activation $f_k$, respectively. Thus Eq. equation 1 defines a form of activation function as a linear combination of a set of basic parameterized non-linear activation functions $f_k(z, \boldsymbol{\beta}_k)$ with the same input $x$ to the neuron. Normally, we have $0 \leq \alpha_{i,k} \leq 1$ for all $k$ and $i$. This setting will take advantage of the low computational costs of existing activation functions, while it will be much easier to implement weights normalization when we need a bounded activation function.

Since the specific activation function corresponding to each neuron only depends on its own activation parameters, the back propagation of these activation parameters by stochastic gradient descent

can be done as follows:

$$\alpha_{ik} \to \alpha_{ik} - \gamma \frac{\partial L}{\partial \alpha_{ik}} \to \alpha_{ik} - \gamma \frac{\partial L}{\partial o_i} \cdot \frac{\partial o_i}{\partial \alpha_{ik}} = \alpha_{ik} - \gamma \frac{\partial L}{\partial o_i} \cdot f_{ik}(z, \boldsymbol{\beta}_{ik})$$

$$\boldsymbol{\beta}_{ik} \to \boldsymbol{\beta}_{ik} - \gamma \frac{\partial L}{\partial \boldsymbol{\beta}_{ik}} \to \boldsymbol{\beta}_{ik} - \gamma \frac{\partial L}{\partial o_i} \cdot \frac{\partial o_i}{\partial \boldsymbol{\beta}_{ik}} = \boldsymbol{\beta}_{ik} - \gamma \frac{\partial L}{\partial o_i} \cdot \alpha_{ik} \frac{\partial f_{ik}(z, \boldsymbol{\beta}_{ik})}{\partial \boldsymbol{\beta}_{ik}} \quad (2)$$

where $i$ is the index of the hidden neuron with output $o_i$ and $k$ is the index of combined flexible activation functions. Here we use a simplified expression that does not include the indices of layer and training examples in each mini-batch. $\gamma$ is the learning rate of gradient descent for all the parameters in activation functions. With the gradients given by $\partial L/\partial \alpha_{ik}$, adaptive optimizers such as AdaGrad (Duchi et al., 2011), Adam (Kingma and Ba, 2014) and RMSProp (Tieleman and Hinton, 2017) can also be applied. In general, gradient descent approach and its derivatives can push the activation parameters toward the direction that minimizes the empirical risk of the model on training data. In practice, considering the different nature of basic model parameters such as weights and biases and activation function parameters, it could be more appropriate to implement different learning rates for each of them. However, this will increase the load of hyper-parameter searching.

To build effective combinations with the general form given by Eq. equation 1, we introduce the following three principles for selecting the components:

- **Principle 1**: Each component should have the same domain as the baseline activation function.
- **Principle 2**: Each component should have an equal range as the baseline activation function.
- **Principle 3**: Each component activation functions should be expressively independent of other component functions with the following definition.

**Definition 1**: If a component activation function $f_k$ is **expressively independent** of a set of other component functions: $f_1, ..., f_n$, there does not exist a set of combination coefficients $\alpha_1, ..., \alpha_n$, inner activation parameters $\beta_1, ... \beta_n$, parameters of the previous linear layers $\boldsymbol{W}'$, $\boldsymbol{b}'$ such that for any input $X$, activation parameters $\boldsymbol{\beta_k}$, and parameters of the previous linear layer $\boldsymbol{W_k}$, $\boldsymbol{b_k}$, the following equation holds:

$$f_k(z_k, \boldsymbol{\beta_k}) = f_k(\boldsymbol{W_k}X + \boldsymbol{b}_k, \boldsymbol{\beta_k}) = \sum_{i=1}^{n} \alpha_i f_i(\boldsymbol{W}'X + \boldsymbol{b}', \boldsymbol{\beta_i}) = \sum_{i=1}^{n} \alpha_i f_i(z', \boldsymbol{\beta_i}) \quad (3)$$

**Proposition 1**: For a single-layer network with $m$ neurons, if a component activation function $f_k$, which is not expressively independent of other components, is excluded, we need at most $2m$ neurons to express the same mapping.

The first two principles are aiming at keeping the same ranges and domains of the information flow with the mapping in each layer. The third principle is aiming at reducing the redundant parameters that do not contribute to the model expressiveness even with limited number of units. A short proof of Theorem is provided in Appendix A.1. For example, $\sigma_1(z) = 1/(1 + e^{-\beta z})$ is not expressively independent with $\sigma(z) = 1/(1 + e^{-z})$ since when $\boldsymbol{W}' = \beta \boldsymbol{W}$, we have $\sigma(\boldsymbol{W}'X) = \sigma_1(\boldsymbol{W}X)$. Therefore, the combined activation $a(z, \beta) = \alpha_1 \sigma(z) + (1 - \alpha_1)\sigma(\beta z)$ will not be a good choice. Based on this, we can then design the combined trainable activation functions for both bounded or unbounded domains.

## 2.2 SIGMOID/TANH FUNCTION EXTENSION FOR RNNS

Sigmoid and Tanh activation functions are widely used in recurrent neural networks, including basic recurrent nets and recurrent nets with cell structure such as LSTMs and GRUs (Gers et al., 1999; Jozefowicz et al., 2015; Goodfellow et al., 2016). For example, an LSTM cell has the functional mapping as follows:

$$\begin{aligned}
\boldsymbol{f}_t &= \sigma(\boldsymbol{W}_{fx}\boldsymbol{x}_t + \boldsymbol{W}_{fh}\boldsymbol{h}_{t-1} + \boldsymbol{b}_f) \\
\boldsymbol{i}_t &= \sigma(\boldsymbol{W}_{ix}\boldsymbol{x}_t + \boldsymbol{W}_{ih}\boldsymbol{h}_{t-1} + \boldsymbol{b}_i) \\
\boldsymbol{o}_t &= \sigma(\boldsymbol{W}_{ox}\boldsymbol{x}_t + \boldsymbol{W}_{oh}\boldsymbol{h}_{t-1} + \boldsymbol{b}_o) \\
\boldsymbol{g}_t &= \tanh(\boldsymbol{W}_{gx}\boldsymbol{x}_t + \boldsymbol{W}_{gh}\boldsymbol{h}_{t-1} + \boldsymbol{b}_g) \\
\boldsymbol{c}_t &= \boldsymbol{f}_t * \boldsymbol{c}_{t-1} + \boldsymbol{i}_t * \boldsymbol{g}_t \\
\boldsymbol{h}_t &= \boldsymbol{o}_t * \tanh(\boldsymbol{c}_t)
\end{aligned} \quad (4)$$

The cell structure includes multiple sigmoid and tanh activation functions, which can be replaced by weighted flexible combination between the original one and another activation function with the same domain. For the sigmoid function, the output should be in the domain of $[0, 1]$, while for tanh the output should be in $[-1, 1]$. In the first case, one simple choice is:

$$o(z; \alpha, \beta) = \alpha \cdot \sigma(z) + (1 - \alpha) \cdot f(z; \boldsymbol{\beta}) \tag{5}$$

where $0 \leq \alpha \leq 1$ and

$$f(z; \boldsymbol{\beta}) = \begin{cases} 0 & \text{if } z < -\frac{1}{2\beta} \\ \beta z + \frac{1}{2} & \text{if } -\frac{1}{2\beta} \leq z \leq \frac{1}{2\beta} \\ 1 & \text{if } z > \frac{1}{2\beta} \end{cases} \tag{6}$$

In Eq. equation 5, $f(z; \beta)$ can be considered as a combination of two Ramp functions bounded between 0 and 1 with parameter $b$. The shapes of a sample of combined activation with Eq. 5 are shown in Appendix A.3. Similarly, we can build a function with the same boundary as tanh function, and use the corresponding combination to replace tanh in the LSTM cell. Consequently, for each combined flexible activation function, there are two parameters to be optimized during the training process. By combining the original activation function $\sigma$ with another function $f(z; \beta)$ with the same boundary using normalized weights, the model can be trained with flexible gates and have the potential to achieve better generalization performance.

## 2.3 ReLu Function Extension for MLPs and CNNs

The outputs of ReLu function is unbounded on positive side, while the derivative with respect to the inputs is a Heaviside step function. To build more flexible activation in the condition when ReLu function is used, we can make a weighted combination between Relu and other non-linear functions with unbounded ends. In the simplest case, we can make a weighted linear combination between ReLu function, $z^3$ and $z^{1/3}$, which can be written as:

$$o(z; \alpha_1, \alpha_2) = \alpha_1 \text{ReLu}(z) + \alpha_2 z^3 + (1 - \alpha_1 - \alpha_2) z^{1/3} \tag{7}$$

where $\alpha$ and $\beta$ are two parameters to be learned with back propagation. By merging basic Relu activation and two functions that are expressively independent with other components, such as $z^3$ and $z^{1/3}$, the model could have the potential to learn non-convexity with much less hidden units. In addition, for very deep networks, the combination of ReLu-like function and smooth non-ReLu function could facilitate the information propagation in considering the Edge of Chaos (EOC) (Hayou et al., 2019).

## 2.4 Layer-wise Regularisation for Activation Parameters

Similar to the weights decay regularisation for model weights in NN models, we introduce regularisation terms for parameters in activation functions to avoid over-parameterization during learning process. When we set the summation of each component's weights in each flexible activation function to 1, it is not suitable to implement a L1 or L2 norm on the absolute value of activation weights. Instead, we use the L2 norm for the absolute difference between each specific activation parameter and the mean of corresponding parameters in the same layer. In addition, we introduce another L2 regularisation term controlling the difference between the trained parameters and the initial parameters of benchmark activation function. This can make sure that the benchmark is actually a specific case of flexible activation, while the variations can be learned to adapt to the training dataset and controlled by these regularisation effects. Thus, the cost function can be written as follows:

$$\begin{aligned} \text{Cost function} =& \text{Predictive Loss} + \delta_1 \sum_j \frac{\lambda_j}{2m_j} \sum_i \sum_k ||\alpha_{ijk} - \bar{\alpha}_{jk}||^2 \\ &+ \delta_2 \sum_i \sum_j \sum_k ||\alpha_{k0} - \alpha_{ijk}||^2 + \text{other terms} \end{aligned} \tag{8}$$

where $\alpha_{ijk}$ refers to the $k$th activation parameter $\alpha_k$ for $i$th element in $j$th layer, $\bar{\alpha}_{jk}$ is the average value of $\alpha_k$ in $j$th layer, $\alpha_{k0}$ is the combination coefficient of $k$th component in basic or standard activation functions (e.g. ReLu), $m_j$ is the number of neurons in $j$th layer, while $\lambda_j$ is the layer-wise regularisation coefficients, and $\delta$ is mutual coefficient. We can consider these two regularization

terms as prioris. For the first one, since in the layer structure of deep neural networks, usually different layer is learning different level of patterns, which could be in favor of using similar activation functions in each layer. Meanwhile, the second regularization terms can be considered as another priori in assuming that the initial activation functions are good enough and the learned activation parameters should not differ too much from the initial values.

## 3 EXPERIMENTS

All the experiments were conducted in the environment of Pytorch 1.3.1, we implemented the embedded functions of sigmoid, relu and prelu in the baseline models and manually created the proposed flexible functions with backward path in the flexible models. The first and second experiments are conducted with a cloud Intel Xeon 8-Core CPU, the third experiment is conducted with a cloud Tesla K80 GPU.

### 3.1 EXPERIMENT WITH RECURRENT NEURAL NETWORKS

For testing the performance of the model with flexible activation in recurrent neural networks, we build a multiple-layer LSTM model. We change the three sigmoid functions in Eq. equation 4 to the parameterized combined function as shown in Eq. equation 5, then compare the model performances in the cases with or without flexible activations.

The dataset being experimented on is a combination of daily stock returns of five G20 countries including Brazil, Canada, India, China and Japan from 02 Jan, 2009, which is a multi-variate time series data. The five returns of each day can be considered as an input vector to the corresponding hidden unit, while the output is one-step ahead forecast given a sequence of historical data. Instead of using random sampling, we directly split the set of sequences with 10 lagging vectors into training set (64%), validation set (16%) and test set (20%), while the learning curve on validation set can be obtained during the training. The loss is selected as the average of mean squared errors of 5 forecasted values with respect to the true values for each example. For the hyper-parameter setting, the batch size was set to be 50, the window size is 10 time steps, the number of epochs is 30, while the optimizer implemented in training is Adam optimizer with the same learning rate on both weights, bias and activation parameters. The initialization of the flexible activation parameters in replacing sigmoid function is $\alpha = 1$ and $\beta = 0.1$, which means that we train them from baseline settings. Four stacked LSTM models with different layer configurations are implemented, then we compare the validation and test performances of these models with fixed and flexible activations by 100 trials with different random initializations for each of them.

We introduce the regularizer proposed in Section 2.4 and tune the corresponding weight decay coefficient $\delta$ for flexible activation parameters. First, for each configuration of layer size, we search for the optimal values of learning rates from 1000 random samples in the range of $[0.001, 0.2]$ with logarithm scale when the fixed activation functions are used. Based on these optimized learning rates for fixed models, we further search for the optimal regularization coefficients for flexible activation functions from the logarithm sale of range $[0.001, 0.1]$ with 30 random samples. The optimal values of the learning rates and the regularization coefficients for activation parameters are listed in Table 1. Here we do provide benefit to fixed models by using their optimal learning rates in the corresponding

Table 1: Summary of hyper-parameter settings for LSTMs in the experiment

| Model | Layer size | Params | Learning rates | Regularization (act) |
|---|---|---|---|---|
| LSTM-1 | [5, 16] | 1408 | 6.71E-3 | 2.50E-1 |
| LSTM-2 | [5, 8, 4, 4] | 800 | 3.47E-3 | 2.50E-1 |
| LSTM-3 | [5, 16, 8] | 2208 | 1.93E-2 | 2.50E-1 |
| LSTM-4 | [5, 16, 16] | 3520 | 4.32E-3 | 2.50E-1 |

flexible models. Meanwhile, to avoid too much extra computational cost paid to flexible models, we use the optimized activation regularization coefficients of LSTM-1 to all the models. Moreover, we use different configurations of layer size to compare models with different capacity to some level.

As is shown in Figure 1, the learning curves of model with flexible activation functions (denoted as "flexible models") on validation set lie below the corresponding curves of models with fixed

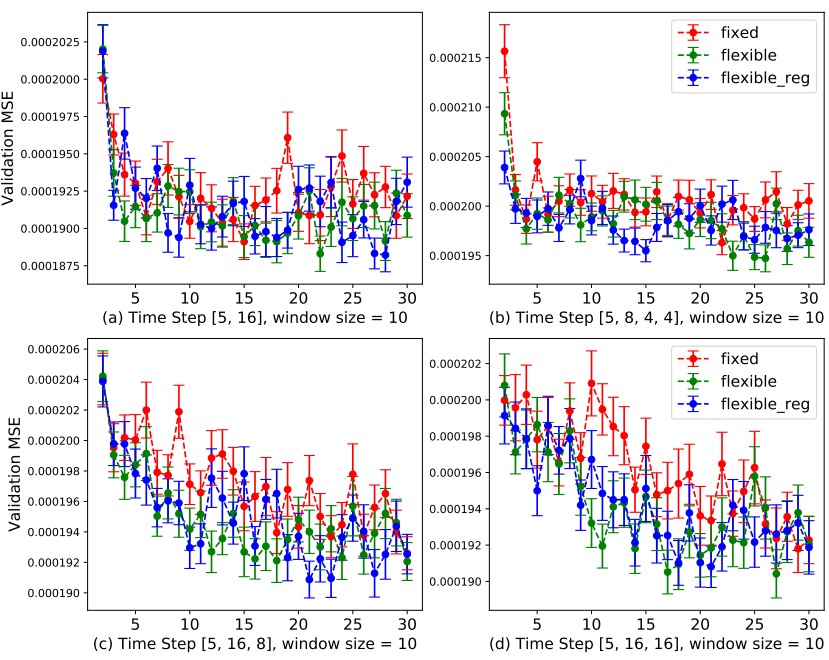

Figure 1: Comparison between the average learning curves (with error bars) of LSTM models with and without regularized flexible activation functions on Multi stock indices return data in forecasting multi-variate return. (a) Two-layer LSTM model with layer sizes: $[5, 16, 8]$; (b) Three-layer LSTM model with layer sizes: $[5, 8, 4, 4]$; (c) One-layer LSTM model with layer sizes: $[5, 16]$; (d) Two-layer LSTM model with layer sizes: $[5, 16, 16]$.

activation functions (denoted as "fixed models") during most of learning time. Still we consider the average of minimum validation loss during the learning process in each configuration. The descriptive statistical analysis of 100 trials in each setting is shown in Table 2.

Table 2: Summary table of stock indices forecasting with Stacked LSTM

| Model | Data | Fixed | Flexible | Flexible (Regularized) |
|---|---|---|---|---|
| LSTM-1 | validation | 1.747E-4 (1.7E-7) | 1.747E-4 (1.5E-7) | **1.746E-4 (1.4E-7)** |
| | test | 7.954E-5 (8.1E-7) | 7.872E-5 (8.1E-7) | **7.724E-5 (4.8E-7)** |
| LSTM-2 | validation | 1.823E-4 (1.9E-7) | **1.804E-4 (2.8E-7)** | 1.809E-4 (3.1E-7) |
| | test | 8.092E-5 (7.7E-7) | **7.811E-5 (5.8E-7)** | 7.841E-5 (5.3E-7) |
| LSTM-3 | validation | 1.780E-4 (2.5E-7) | 1.771E-4 (2.3E-7) | **1.770E-4 (2.2E-7)** |
| | test | 8.046E-5 (9.3E-7) | 7.939E-5 (6.2E-7) | **7.794E-5 (5.1E-7)** |
| LSTM-4 | validation | 1.771E-4 (2.1E-7) | **1.756E-4 (2.5E-7)** | 1.757E-4 (1.9E-7) |
| | test | 7.925E-5 (7.0E-7) | 7.920E-5 (6.9E-7) | **7.919E-5 (5.9E-7)** |

We can see that in all the configurations of layer size, flexible models outperforms fixed models in terms of minimum validation error and the test error. Especially in the best-performed case with layer size of $[5, 16]$, which is neither the largest or the smallest in terms of the number of parameters, the regularized flexible model achieves an average minimum validation error of 1.746E-04, significantly outperforms all the other models evaluated in this experiment. Further pair-wise statistical tests with normal assumption give p-values between $10^{-2} \sim 10^{-6}$. Meanwhile, this model has only

6.82% more parameters compared with the corresponding fixed model as is calculated in A.3, which is still much smaller than the poorly performed ones with larger sizes in this experiment. Moreover, this performance improvement can also be observed in further experiments on other combinations of stock indices. Another randomly drawn combination is investigated in A.4.2.

## 3.2 EXPERIMENT WITH DEEP AUTO-ENCODER

The deep auto-encoder based on neural networks is widely implemented in data compression and dimension reduction (Baldi, 2012; Goodfellow et al., 2016). In this experiment, we use two fully connected auto-encoder networks, for which both the encoder and decoder have three hidden layers. The difference between these two baseline models is the sizes of two layers in each of the encoder and decoder. The following are the flow graphs of these two models.

$$\text{Input(28*28)} \rightarrow \text{Linear(28*28, 128)} \xrightarrow{\text{ReLu}} \text{Linear(128, } d_1) \xrightarrow{\text{ReLu}} \text{Linear(}d_1\text{, 12)}$$

$$\xrightarrow{\text{ReLu}} \text{Linear(12, } d_2) \rightarrow \text{Coding} \rightarrow \text{Linear(}d_2\text{, 12)} \xrightarrow{\text{ReLu}} \text{Linear(12, 64)} \xrightarrow{\text{ReLu}} \quad (9)$$

$$\text{Linear(64, 128)} \xrightarrow{\text{ReLu}} \text{Linear(128, 28*28)} \rightarrow \text{Output(28*28)}$$

Here we use $d_1$ and $d_2$ to denote the layer sizes differ in two models. For the first model "AE1", $d_1 = 36$ and $d_2 = 6$, while for the second model "AE2", $d_1 = 64$ and $d_2 = 3$. For models with flexible activation, we replace the ReLu activation function with the parameterized function shown in Eq.equation 9, as well as the existing PReLu activation (He et al., 2015). To avoid adding too many extra parameters, we only introduce flexible activation functions in the 3th and 4th ReLu layes in AE1, while for AE2, they are only introduced in the 2th and 5th ReLu layers. In each trial, we

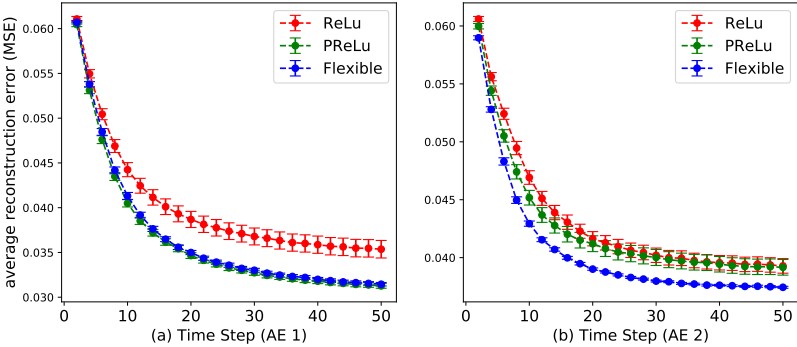

Figure 2: Comparison between the average learning curves (with error bars) of auto-encoder models with and without regularized flexible activation functions on MNIST dataset.

randomly sampled 4,800 training examples, other 1,200 validation examples from 60,000 training examples in the original MNIST dataset, and evaluate the model by the whole 10,000 test examples. The batch size was set to 100 and the learning rates of Adam are optimized based on the validation performance of the baseline models, which is set to be 0.00645. The training curves are averaged by 50 trials, and the results is demonstrated in Figure 2. The hyper-parameter searching procedure is the same as that in Section 3.1. The optimal learning rates of fixed models is 0.00645, and the regularisation coefficients of flexible activation functions is set to be 0.1. Based on the optimized hyper-parameters of each model, 50 experiments are launched for comparing the model with fixed and flexible activations, and the results is demonstrated in Figure 2.

As we can see, for both the two auto-encoder architectures, flexible models out-perform the fixed ones with stable performances. In AE2, the newly proposed activation function significantly out-perform PReLu in almost all the epochs, where the flexible activations are added in 2th and 5th ReLu layers. Table 3 gives the corresponding summary for comparing both the validation and test performances of these configurations. It is shown that the test cost and minimum validation cost of flexible models are generally better than that of fixed models with statistical significance in both the

Table 3: Comparison of auto-encoder models with and without flexible activation functions

| Model | Params | Data | ReLu | PReLu | Flexible |
|-------|--------|------|------|-------|----------|
| AE1 | 212,070 | validation | 3.511E-2 (9.7E-4) | **3.106E-2 (2.6E-4)** | 3.120E-2 (1.9E-4) |
| | | test | 3.519E-2 (9.9E-4) | **3.108E-2 (2.7E-4)** | 3.122E-2 (1.9E-4) |
| AE2 | 219,891 | validation | 3.898E-2 (6.6E-4) | 3.877E-2 (6.6E-5) | **3.708E-2 (7.8E-5)** |
| | | test | 3.960E-2 (6.5E-3) | 3.948E-2 (6.9E-5) | **3.770E-2 (6.9E-5)** |

two stacked auto-encoder architectures. This advantage can be justified by the results on the test set. The performance of AE1 is generally better since the length of encoded vector is 6 rather than 3 as in AE2. Meanwhile, the flexible models has only less than 0.03% extra amount of parameters compared with the corresponding fixed ones. Further experiments demonstrated in A.4.3 show that flexible auto-encoder models with 5 encoder layers and 5 decoder layers also outperform fixed models.

## 3.3 EXPERIMENTS WITH CONVOLUTIONAL NEURAL NETWORK (LENET-5)

This experiment investigates the performance of layer-wisely regularized PRelu function in CNNs, it is done with LeNet-5 on CIFAR-10 for image classification (LeCun et al., 2015b; He et al., 2015). The model architecture is shown as follows:

$$
\text{Input}(32*32*3) \rightarrow \text{Conv2d}(3, 6, 5) \xrightarrow{\text{ReLu}} \text{MP}(2) \rightarrow \text{Conv2d}(6, 16, 5) \xrightarrow{\text{ReLu}} \text{MP}(2)
$$
$$
\rightarrow \text{Linear}(16*5*5, 120) \xrightarrow{\text{BN, ReLu}} \text{Linear}(120, 84) \xrightarrow{\text{BN, ReLu}} \text{Linear}(84, 10) \rightarrow \text{Output} \tag{10}
$$

where "MP" refers to max-pooling layer, while "BN" refers to batch normalization. In flexible models , we only replace the ReLu activation in the last layer with equation 6. In each trial, we still randomly sample 20% of the training set in CIFAR-10 as validation set and use the whole remaining 80% as the training data. The whole test set in CIFAR-10 are used as the test set in our experiment as well. With a naive random search, the optimized learning rate for fixed model is 0.0012, while the regularization coefficients for the parameter based on this learning rate in flexible is 0.032. Meanwhile, the batch size is still set to be 100.

The average cross-entropy loss on validation set during the training whole time of 10 epochs is given by Figure 3. We can see that the average loss curve of PReLu with layer-wise regularization is almost always below the curve of fixed ReLu and PReLu without regularization, while all the three sets of models are over-fitted after 8th epoch. To check the significance, the corresponding summary statistics for 50 trials is shown in Table 4, where the test results are given by accuracy in classification.

We can see that in Table 4, the average minimal validation cross-entropy loss of regularized flexible models still achieves significant improvement compared with fixed ones (ReLu) and flexible model without regularization (PReLu). Even though the mean accuracy of fixed model seems to be slightly better with batch normalization, it is quite insignificant considering the standard errors. Since the stopping time (10 epochs) is not optimized for both models, this does not make much sense considering the existence of over-fitting. The significant validation results indicate that layer-wise regularization on activation parameters could provide an improvement on models with flexible activation functions even in finely designed benchmark architectures such as LeNet-5.

Table 4: Comparison of ReLu, PReLu and PReLu with Layer-wise Regularization in LeNet-5

| Model | Params | Data | ReLu | PReLu | PReLu_reg |
|-------|--------|------|------|-------|-----------|
| LeNet-5 | 61,706 | validation | 1.084 (5.7E-3) | 1.0796 (4.6E-3) | **1.075 (4.7E-3)** |
| | | test (acc) | 62.35 (2.0E-1) | 62.39 (1.7E-1) | **62.67 (1.6E-1)** |
| LeNet-5 (BN) | 62,522 | validation | 1.086 (5.9E-3) | 1.083 (4.9E-3) | **1.072 (4.9E-3)** |
| | | test (acc) | **62.53 (2.3E-1)** | 62.36 (2.0E-1) | 62.48 (1.7E-1) |

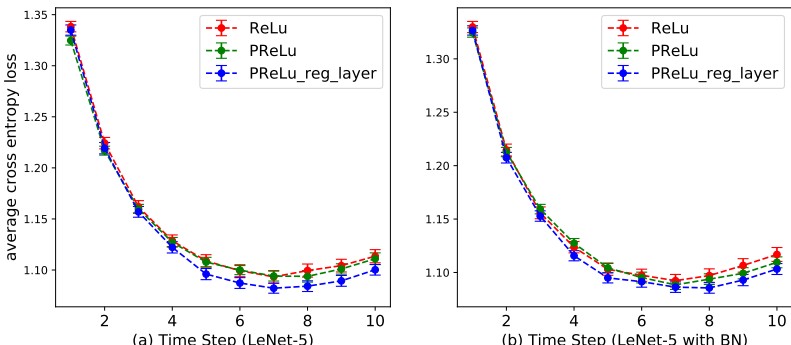

Figure 3: Comparison between the average learning curves (with error bars) of CNN models (LeNet-5) with and without regularized flexible activation functions on CIFAR-10 dataset.

## 4    CONCLUSION

In this study, we proposed a set of principles for designing flexible activation functions in a weighted combination form. Especially, we developed a novel flexible activation function that can be implemented to replace sigmoid and tanh functions in the RNN cells with bounded domains, as well as an alternative one to replace ReLu or PReLu activation in architectures with unbounded domains. In addition, two regularization terms considering the nature of layer-wise feature extraction and goodness of original activation functions are proposed, which is essential in achieving stable improvement of the models. Experiments on multiple time series forecasting show that, with replacing sigmoid activation by the flexible combination proposed in this study, stacked LSTMs can achieve significant improvement. Meanwhile, another proposed flexible combination could significantly improve the performance of auto-encoder networks in image compression. Further experiments indicate that the models with moderately optimized regularized coefficients could also improve the performance of PReLu in CNN (LeNet-5) architectures for image classification. In future studies, it is worthwhile to investigate other flexible activations in combined form based on the proposed framework and principles in this paper, while theoretical justification of the goodness or effectiveness of these activation functions is also a topic of interest.

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

# A APPENDIX

## A.1 EXPRESSIVELY INDEPENDENT

**Definition 1**: If a component activation function $f_k$ is **expressively independent** of a set of other component functions: $f_1, ..., f_n$, there does not exist a set of combination coefficients $\alpha_1, ..., \alpha_n$, inner activation parameters $\beta_1, ... \beta_n$, parameters of the previous linear layers $W'$, $b'$, such that for any input $X$, activation parameters of function $\beta_k$, and parameters of the previous linear layer: $W_k$, $b_k$, the following equation holds:

$$f_k(W_k X + b_k, \beta_k) = \sum_{i=1}^{n} \alpha_i f_i(W' X + b', \beta_i) \tag{11}$$

**Proposition 1**: For a single-layer network with $m$ neurons, if a component activation function $f_k$, which is not expressively independent of other component functions, is excluded, we need at most $2m$ neurons to express the same mapping.

**Proof**: For any original combined activation function: $F = \sum_{i=1}^{n} \alpha_i f_i$. Assume that $f_k$ is not expressively independent. If we exclude $f_k$ from the combination and get $F' = \sum_{i=1, i \neq k}^{n} \alpha_i f_i$, then for any input $X$, parameters of the previous linear layer: $W$, $b$, and activation parameters: $\{\beta\}_{i=1}^{n}$ of a specific neuron, we have:

$$F(WX + b, \beta) = \sum_{i=1, i \neq k}^{n} \alpha_i f_i(WX + b, \beta_i) + f_k(WX + b, \beta_k) \tag{12}$$

and there exist $W'$, $b'$ and $\{\beta\}_{i=1, i \neq k}^{n}$ such that:

$$F(WX + b, \beta) = \sum_{i=1, i \neq k}^{n} \alpha_i f_i(WX + b, \beta_i) + \sum_{i=1, i \neq k}^{n} \alpha_i' f_i(W' X + b', \beta_i') \tag{13}$$

which can be expressed by two neurons with function $\{f\}_{i=1, i \neq k}^{n}$ with corresponding weights and bias in the previous linear layer. Therefore, for a single-layer network with $m$ neurons, we need at most $2m$ neurons without $f_k$ to express the any mappings by the original function.

## A.2 THE SHAPES OF COMBINED FLEXIBLE ACTIVATION FUNCTIONS

## A.3 CALCULATING THE NUMBER OF PARAMETERS

When batch-normalization is not implemented, the number of parameters in each deep neural network is equal to the number of weighs plus the number of bias. For an FFNN, $N = N_W + N_b = \Sigma_{i=1}^{L-1} n_i n_{i+1} + \Sigma_{i=2}^{L} n_i$, where $L$ is the number of layers including input layer, hidden layer and output layer, while $n_i$ is the number of units in $i$th layer. If flexible activations are introduced in all hidden and output units, with two independent extra parameters in each unit, the total number of model parameters will increase by $N_a = 2 \sum_{i=2}^{L} n_i$. The number of parameters in basic RNN models is depend on the type of hidden cells used, in general it is $N = g \sum_{i=1}^{L-1} (n_{i+1}(n_{i+1} + n_i) + n_{i+1})$, where $g$ is the number of weight matrices in each cell. For basic RNN, $g = 1$, for GRU, $g = 3$, and for LSTM, $g = 4$. Meanwhile, the extra number of parameters needed for RNNs with flexible activations is $N_a = 2 \sum_{i=2}^{K} s_i n_i$, where $s_i$ is the number of activation functions replaced by flexible ones in $i$th layer.

Following this derivation, in the LSTM models used in Section 3, we can make a summary table for the increasing ratio of model parameters in each case. As shown in Table 5, we can see that the models with flexible activations have about 5% to 10% increase in the number of parameters. However, it is shown in Table 6 that model 1, model 3 and model 4 have very similar validation performances during training even though their number of parameters varies in much higher (even more than 100%) proportions, while the corresponding validation performances for models with flexible activations are better than all the models with fixed activations.

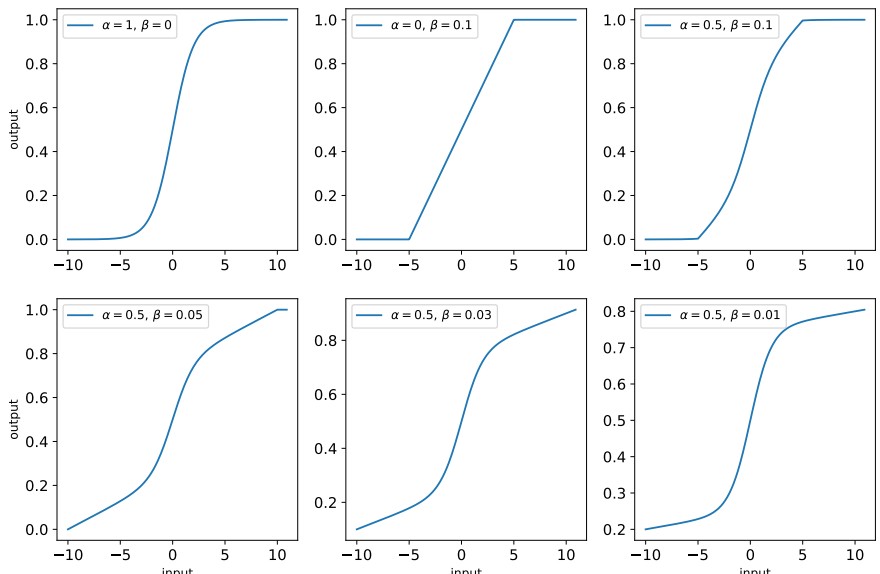

Figure 4: The shapes of combined activation function proposed in Section 2.2 with different set of parameters.

Table 5: Comparison of number of parameters in different settings of LSTM models.

| model | layer size | basic model | flexible activation | ratio of increase |
|-------|-----------|-------------|--------------------|--------------------|
| 1 | [5, 16, 8] | 2208 | 144 | 6.52% |
| 2 | [5, 8, 4, 4] | 800 | 96 | 12.00% |
| 3 | [5, 16] | 1408 | 96 | 6.82% |
| 4 | [5, 16, 16] | 3520 | 192 | 5.45% |

Similarly, in Table 2, the best performed model with layer size $[5, 18]$ has a single hidden layer and the number of parameters is between the model with layer sizes $[5, 16, 8]$ and $[5, 8, 4, 4]$. Therefore, the number of parameters does not really matters in this range when we consider the minimum validation performance in a long training time, and the improvement of model performance can be explained by the advantage of introducing flexible activation with trainable parameters.

For the two deep auto-encoder models studied in Section 3.3, the number of basic model parameters can be calculated with the formula as same as feed-forward neural network, which is: $N = N_W + N_b = \Sigma_{i=1}^{L-1} n_i n_{i+1} + \Sigma_{i=2}^{L} n_i$. Based on the layer structure shown in Eq. equation 9, the deep auto-encoder model AE2, for example, has 219,891 basic parameters. Meanwhile, the extra parameters in flexible activation functions is 48, which is only 0.022% of the basic parameters.

For the CNN Lenet-5 model in Section 3.3, the basic model parameters include those in convolutional layers as well as those in full connected layers. The number of parameters of each convolutional neural network is: $N_c = (n * n * l + 1) * k$, where $n$ is the filter size, while $l$ and $k$ represent the input and output feature maps. The total number of two convolutional layers in Eq. equation 10 is 2,572, and the number of parameters in fully connected layers is 59,134, giving a total number of 61,706 parameters in our Lenet-5 model. For model with batch normalization, the number of batch normalization parameters is 4*(120+84) = 816, and the total number of parameters is $61,706 + 816 = 62,522$. In addition, for flexible models, the extra activation parameters introduced

into the second last layer is 84*2=168, which is only about 0.13% of the total number of basic parameters.

To sum up, the results at least show that we can achieve a quite significant performance improvement with only a small proportion of extra parameters in the flexible activation functions. Although the same mapping may be learned without flexible activation functions, it may need much more number of parameters or much larger effort of hyper-parameter searching with a bunch of different model architectures.

### A.4 Extended Experimental Results

#### A.4.1 Forecasting uni-variate target without regularisation on activation parameters

In the first experiment for LSTM model on multi-variate financial time series data, we focus on the case when the target is uni-variate. Here we choose the one-step-ahead daily return of Brazil index as the target to be predicted, while the indices of other four countries in the dataset are used as predictors along with the historical data of the target. To avoid the uncertainty caused by different hyper-parameters for different models, we use the recommended learning rate of 0.01 for Adam optimizer for all the models in this experiment. In addition no regularizer for flexible activations is added in this stage, which will be considered later.

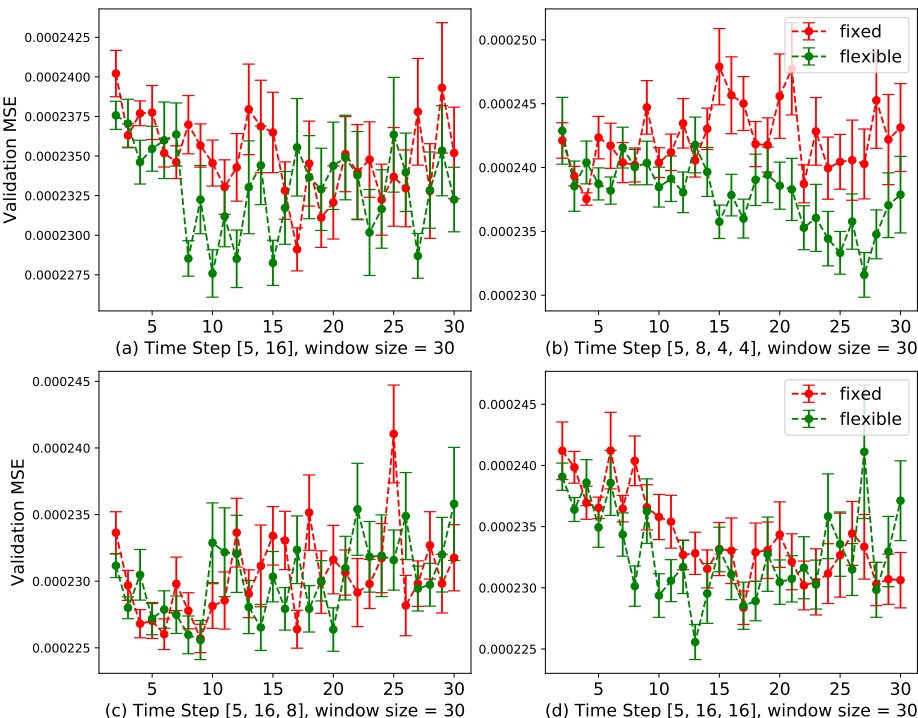

Figure 5: Comparison between the learning curve of LSTM model with and without flexible activation functions on Multi stock indices return data in forecasting uni-variate return. (a) Two-layer LSTM model with layer sizes: $[5, 16, 8]$; (b) Three-layer LSTM model with layer sizes: $[5, 8, 4, 4]$; (c) One-layer LSTM model with layer sizes: $[5, 16]$; (d) Two-layer LSTM model with layer sizes: $[5, 16, 16]$.

The result for learning curve of recurrent neural networks for multiple time series stock indices data is shown in Figure 5. We find that for all stacked LSTM models with different layer configurations, the version with flexible activations outperforms the version with fixed activation in most of the learning time. To understand the significance of the improvement, we perform two statistical tests by assuming the validation performance of the model with different random initialization following normal distributions.

Table 6: Summary table of stock indices forecasting with Stacked LSTM

| model | Layer size | Fixed Activation | | Flexible Activation | | p-value |
|-------|-----------|-----------------|------|-------------------|------|---------|
|       |           | mean of min | S.E. | mean of min | S.E. | |
| 1 | [5, 16, 8] | 2.194E-04 | 1.77E-07 | **2.185E-04** | 1.91E-07 | 8.173E-03 |
| 2 | [5, 8, 4, 4] | 2.272E-04 | 9.09E-07 | **2.228E-04** | 7.47E-07 | 8.311E-05 |
| 3 | [5, 16] | 2.191E-04 | 1.77E-07 | **2.186E-04** | 1.91E-07 | 4.507E-02 |
| 4 | [5, 16, 16] | 2.193E-04 | 1.26E-07 | **2.187E-04** | 1.27E-07 | 2.632E-04 |

It is shown in Table 6 that the averages of minimum validation cost by using stacked LSTM with flexible activation are significantly smaller than that with fixed activation in terms of stock indices forecasting. On the other hand, we find that their costs after 30 epochs on test set both outperform traditional ARIMA models in on the same dataset (Asteriou and Hall, 2011; Ariyo et al., 2014), which is 0.0002217. In terms of the standard errors, it does not show that one of these two set of models are more robust to the noise in initialization than the other.

Table 7: Comparison between different forecasting methods on test set

| model | RW | ARIMA(5,1,0) | LSTM fixed | | LSTM flexible | |
|-------|-----|-------------|-----------|------|--------------|------|
|       |     |             | mean | S.E. | mean | S.E. |
| test mse | 3.030E-04 | 2.217E-04 | 1.669E-04 | 3.173E-06 | **1.644E-04** | 2.18E-06 |

Table 7 shows the performances of different forecasting methods on test set, where the stacked LSTM models with flexible activations outperforms other methods including Naive random walk, ARIMA(5, 1, 0) and LSTM with fixed activations, in terms of the average test MSE for the whole 200 trials. However, since the test results of LSTM model is based on the model after 30 epochs rather than the optimal number of epoch during the training, the performance of LSTM with flexible activations is not significantly lower than that of fixed activations considering both the means and standard errors.

Table 8: Summary table of stock indices forecasting with Stacked LSTM

| model | Layer size | Fixed Activation | | Flexible Activation | | p-value |
|-------|-----------|-----------------|------|-------------------|------|---------|
|       |           | mean of min | S.E. | mean of min | S.E. | |
| LSTM-1 | [5, 16, 8] | 1.980E-04 | 1.55E-06 | **1.972E-04** | 2.39E-06 | 2.47E-02 |
| LSTM-2 | [5, 8, 4, 4] | 2.006E-04 | 6.91E-07 | **1.998E-04** | 2.07E-06 | 4.91E-03 |
| LSTM-3 | [5, 16] | 1.916E-04 | 6.39E-07 | **1.913E-04** | 4.89E-07 | 5.03E-03 |
| LSTM-4 | [5, 16, 16] | 1.933E-04 | 1.56E-06 | **1.918E-04** | 1.12E-06 | 3.88E-08 |

A.4.2   LSTM WITH A DIFFERENT COMBINATION OF STOCK INDICES

In the extended experiment on multiple stock indices forecasting with LSTM, we selected a combination of four different stock indices including indices of British, Australia, Korea and Russia. We do the experiment with two different window sizes (10 and 30), while the regularization coefficients for activation parameters are not optimized on the new dataset and window size. The corresponding results are shown in Figure 7 and Table 9.

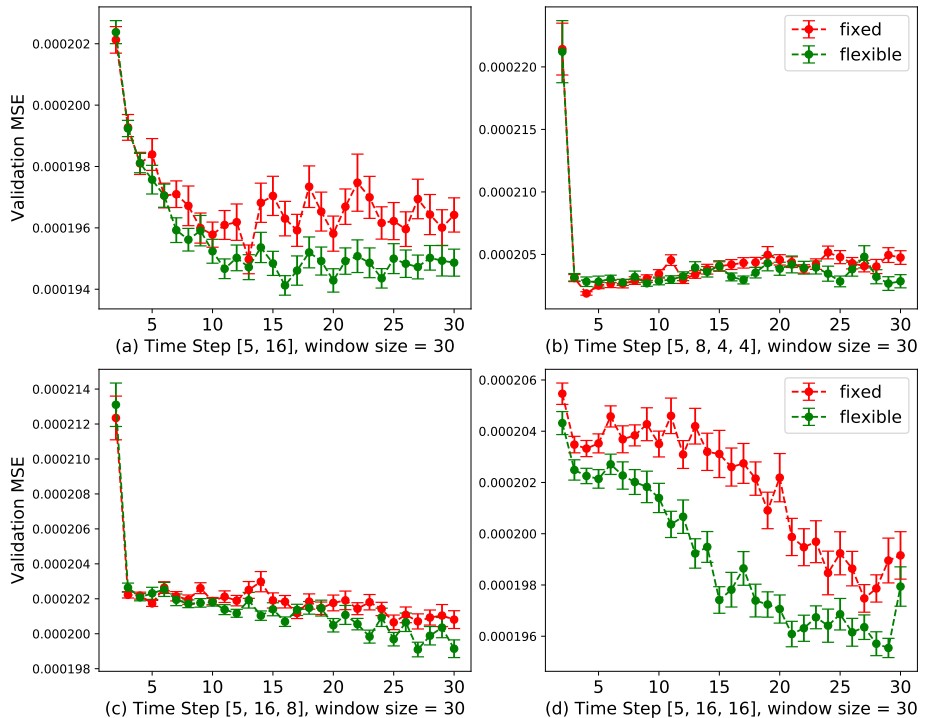

Figure 6: Comparison between the average learning curves (with error bars) of LSTM models with and without regularized flexible activation functions on Multi stock indices return data in forecasting multi-variate return.

Table 9: Summary table of stock indices forecasting with Stacked LSTM

| Model | Window Size | Data | Fixed | Flexible | Flexible (Regularized) |
|---|---|---|---|---|---|
| LSTM-1 | 10 | validation | 1.552E-4 (1.6E-7) | 1.549E-4 (1.4E-7) | **1.549E-4 (1.2E-7)** |
|  | 10 | test | 7.863E-5 (1.0E-6) | 7.871E-5 (1.3E-6) | **7.774E-5 (1.4E-6)** |
| LSTM-1 | 30 | validation | 1.682E-4 (3.3E-8) | 1.692E-4 (3.9E-7) | **1.681E-4** (3.3E-7) |
|  | 30 | test | 7.780E-5 (6.0E-7) | **7.661E-5 (3.1E-7)** | 7.761E-5 (4.9E-7) |

### A.4.3  5-layer Auto-encoder

This is for checking the robust of flexible auto-encoder models with deeper architecture. The diagram of data flow is shown as follows.

$$
\text{Input}(28*28) \rightarrow \text{Linear}(28*28, 128) \xrightarrow{\text{ReLu}} \text{Linear}(128, 64) \xrightarrow{\text{ReLu}} \text{Linear}(64, 36)
$$

$$
\xrightarrow{\text{ReLu}} \text{Linear}(36, 12) \xrightarrow{\text{ReLu}} \text{Linear}(12, 6) \rightarrow \text{Coding} \rightarrow \text{Linear}(6, 12) \xrightarrow{\text{ReLu}} \text{Linear}(12, 36) \xrightarrow{\text{ReLu}}
$$

$$
\text{Linear}(36, 64) \xrightarrow{\text{ReLu}} \text{Linear}(64, 128) \xrightarrow{\text{ReLu}} \text{Linear}(128, 28*28) \rightarrow \text{Output}(28*28)
$$

$$(14)$$

Here we still introduce two models: AE3 and AE4. In AE3, flexible activations are added to replace the 3th and 6th ReLu in Eq. 14, while in AE4, flexible activations are added to replace the 4th and 5th ReLu. The experimental results are shown in Figure 8 and Table 10.

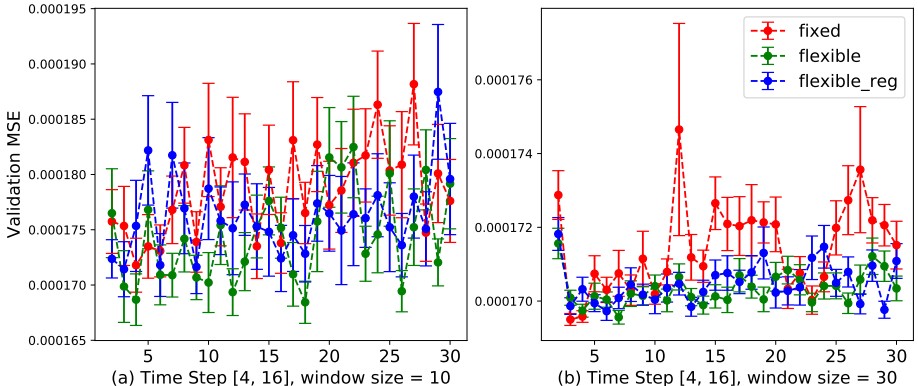

Figure 7: Comparison between the average learning curves (with error bars) of LSTM models with and without regularized flexible activation functions on Multi stock indices return data in forecasting multi-variate return. (a) layer size [4, 16], window size 10; (b) layer size [4, 16], window size 30

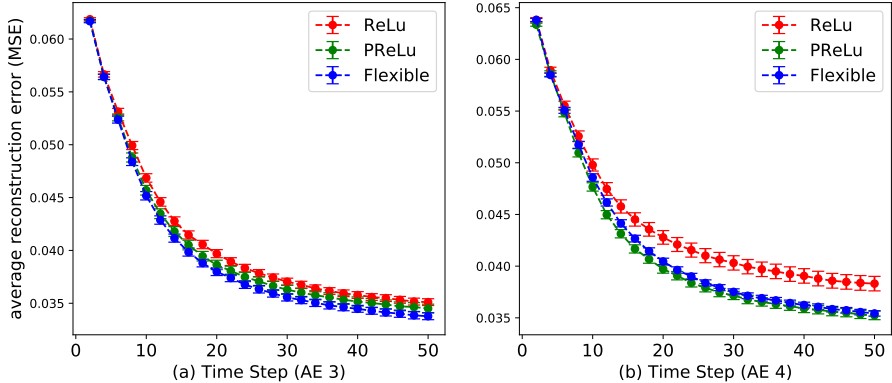

Figure 8: Comparison between the average learning curves (with error bars) of auto-encoder models with and without regularized flexible activation functions on MNIST dataset (5-Layer).

Table 10: Comparison of auto-encoder models (5-layer) with and without flexible activation functions

| Model | Params | Data | ReLu | PReLu | Flexible |
|-------|--------|------|------|-------|----------|
| AE3 | 223,974 | validation | 3.481E-2 (3.1E-4) | 3.427E-2 (4.0E-4) | **3.351E-2 (3.2E-4)** |
|  |  | test | 3.471E-2 (3.3E-4) | 3.408E-2 (4.2E-4) | **3.337E-2 (3.3E-4)** |
| AE4 | 223,974 | validation | 3.799E-2 (7.0E-4) | **3.496E-2 (3.9E-5)** | 3.521 (2.6E-4) |
|  |  | test | 3.744E-2 (7.0E-4) | **3.436E-2 (3.9E-5)** | 3.460E-2 (2.5E-4) |

