# OpenReview forum: "COMBINED FLEXIBLE ACTIVATION FUNCTIONS FOR DEEP NEURAL NETWORKS"
_ICLR.cc/2020/Conference — Reject_

### Official Review · AnonReviewer2 · 2019-10-23
**Official Blind Review #2**

**Rating:** 3

**Review:**

This paper presents a family of parameterized composite activation functions, and a regularization technique for parameterized activation functions in general. While the three sets of experiments show potentially promising results, they aren't able to disambiguate clearly between the effect of the activation function you introduce and the effect of additional parameters in general, or between the effect of the regularization technique you introduce and the effect of regularization in general. I would really like to see the kind of carefully baselined, ablated, and hyperparameter-tuned results that would justify adding the techniques introduced to the toolbox of a typical deep learning practitioner.

Some feedback:

Typos:
- The formatting is off a bit (shifted horizontally?)
- In the abstract: RPeLu -> PReLU
- p. 5: "basement settings"->baseline, "logarithmic sale"->scale
- p. 6: basement->baseline again
- p. 7 etc.: trail->trial

Feedback:
- Intro:
  - Explain why maxout is similar to training piecewise activation functions?
  - It's unclear what "making the most of the non-linear properties by introducing adaptation policy on the input" means
  - An "autoencoder" is not an architecture as much as a broad family of architectures coupled with training approaches (I'm guessing you mean a fully-connected autoencoder)
- Methodology:
  - consider using "f" instead of "a" so it's easier to tell apart from alpha?
- Experiments:
  - I'm not sure I'm convinced by the statistical tests used on the LSTM results; they demonstrate that your approach, with a few specific sets of hyperparameter settings, does better than the baseline, but not that this represents a valid claim about your activation function's effect on this LSTM model in general.
  - The autoencoder experiments are even less convincing, in that they represent two seemingly arbitrary network architectures, with equally arbitrary placement of the activation function in one of them.
  - The regularization experiments use LeNet-5, which is not a compelling benchmark architecture with respect to contemporary practice. The effects of regularization techniques can be very different in different regimes of dataset and network size.
- Code:
  - I'm not sure why you performed backprop by hand for your activation functions and used torch.Function, rather than writing them directly as nn.Modules that make use of PyTorch autograd
  - There are lots of details in the code that aren't in the paper; in general, papers should aim to be relatively self-contained (although I'm very glad to see your code, and it's pretty simple to follow)

**Experience Assessment:**

I have published one or two papers in this area.

**Review Assessment: Checking Correctness Of Derivations And Theory:**

I assessed the sensibility of the derivations and theory.

**Review Assessment: Checking Correctness Of Experiments:**

I carefully checked the experiments.

**Review Assessment: Thoroughness In Paper Reading:**

I read the paper at least twice and used my best judgement in assessing the paper.

---

> ### Author Response · Authors · 2019-11-09
> **Response to reviewer #2**
>
> Thanks very much for your time and feedback.
>
> Review:
> -	We will try to add more experimental results for checking the effects of proposed flexible activation functions by comparing with baseline models.
> Typos:
> -	Thanks for pointing out the typos, we have corrected them in the updated version, which will be uploaded later.
>
> Feedback:
> a.	Intro
> -	In the original paper of maxout (Goodfellow et al. 2013), the output of each hidden unit with maxout activation is the maximum of k affine feature maps. In each piece of input space, it takes the affine function with the largest output value, which lead to a convex piece-wise function that can be used to approximate any convex functions.
> -	In the original paper (Yang et al. 2018), they model the context and the conditional next-token distribution as a softmax function, while the context vector h and word embedding w depend on the input next tokens and context with learnable parameters, which can increase the model capacity to be more adaptive to high rank language models with non-linear properties. We will try to improve the expression in the updated version.
> -	I think we have already used “deep auto-encoder based on neural networks” and “fully connected auto-encoder” in the first paragraph of section 3.2, but shorten it to “auto-encoder” in later context to reduce the length of the paper. The “auto-encoder” in the abstract has been changed to “fully connected auto-encoder” in the updated version.
> b.	Methodology
> -	We will update the notation “a” in the equations to “f”.
> c.	Experiments
> -	In our experiments, we find that in all architectures of multi-layer LSTM being tested, the newly proposed flexible sigmodal activation and its regularized version achieves an improvement, and most of them are statistically significant. As we know, in general the goodness of activation function depends on the learning tasks and the model architectures. Even PRelu cannot outperform the baseline models with Relu in all image classification tasks with CNNs. Thus we think if it can provide a significant improvement for a proportion of tasks and architectures, we can consider it as a discrete hyper-parameter in a list with other promising baseline activation functions, which is much more computational efficient than trying a new architecture with hyper-parameter optimization from scratch.
> -	We know that LeNet-5 is a relatively small model that does not provide competitive results on CIFAR-10. However, considering our limited computational resources, it seems to be computational expensive and time consuming to train large models such as Resnet for a relatively large number of times for hyper-parameter optimization and statistical test. We find most related study on flexible activation functions only perform 5-10 runs without statistical test.
> d.	Code
> -	We define an independent function with backward path considering the possibility for introducing non-differentiable components into our flexible activation functions. In fact, the newly proposed flexible sigmodal activation involves a piecewise function, whose derivative may not be calculated with pytorch.
> -	We will try to make more explanation for the experimental details in the updated version.

---

### Official Review · AnonReviewer1 · 2019-10-23
**Official Blind Review #1**

**Rating:** 1

**Review:**

The authors introduce a parameterized activation function to learn activation functions that have sigmoidal shapes that can be used in LSTMs. The authors apply their method to a dataset of forecasting stocks as well as to CIFAR-10. They also propose a method to regularize the activation function parameters.

The authors propose an activation function that helps very little in certain cases. I am not familar with this stock prediction dataset, but the differences shown are often less than 1%. I am familiar with CIFAR-10 and the architecture they use give bad baseline results and their method of regularizing a previous learned activation function (PReLU) gives marginally better results in half the cases.

There have been many learned activation functions proposed that give significantly better results than these. This paper is simply proposing another one and showing little to no improvement.

**After reading author feedback**
My score stays the same.

There were some issues I had with the response:
"For stock return forecasting, 1% improvement with statistical significance is sufficiently import...As we know, most newly proposed unbounded activation functions (flexible or not) cannot outperform ReLu by more than 0.5% in terms of test accuracy in a large range of image classification tasks."
The 1% improvement was on error, not accuracy, so the two are not related. If referring to error rate, there are many that can make improvements greater than 10% (APLs, SReLUs, Swish, etc.)
If the authors would like to claim superiority to other activation functions, they should compare to them directly.

In addition, my concern about the poor CIFAR-10 baseline was not addressed. The results from the 2014 dropout paper gives significantly better results (Dropout:  A simple way to prevent neural networks from overfitting) than the authors' baseline.


**Experience Assessment:**

I have published one or two papers in this area.

**Review Assessment: Checking Correctness Of Derivations And Theory:**

I did not assess the derivations or theory.

**Review Assessment: Checking Correctness Of Experiments:**

I assessed the sensibility of the experiments.

**Review Assessment: Thoroughness In Paper Reading:**

I made a quick assessment of this paper.

---

> ### Author Response · Authors · 2019-11-09
> **Response to reviewer #1**
>
> Thank you very much for your feedback.
>
> -	For stock return forecasting, 1% improvement with statistical significance is sufficiently import. We will use the newly proposed sigmodal function rather than sigmoid as the activation in LSTMs for any multi-variate time series forecasting tasks. As we know, most newly proposed unbounded activation functions (flexible or not) cannot outperform ReLu by more than 0.5% in terms of test accuracy in a large range of image classification tasks.
> -	In the second experiment, the improvement is quite significant from the baseline ReLu activation. Even though in some cases it can be outperformed by PReLu, we can still consider the choice between them as a hyper-parameter to be selected. The original study on PReLu did not mention its advantage in image compression with densely connected Auto-encoder. In addition, the experiment also demonstrates that the goodness of different activation functions may depend on the layer where we put them.
> -	We use LeNet-5 since we hope to make statistical test with a relatively large number of trials. In fact, we find that most related studies only perform 5 or 10 trials without testing the statistical significance or giving the error bars for each time step. Models like ResNet and DenseNet can provide better validation/test accuracies on CIFAR10, but they require much more computational power for training while still outperformed by the current SOTA achieved by NAS. Therefore, we did not use these large models for testing the significance as we are not focusing on computer vision tasks.
> -	We do not agree with your conclusion that we are simply proposing another activation function showing little to no improvement. Actually, we proposed a family of activation functions with a general form that can be introduced for both bounded and unbounded domains, which is highly extendable. In the first experiment, the discrepancy is small but with statistical significance. As we know, it is probably the first flexible sigmodal activation function introduced to LSTM. In the second experiment, the result is significant and the discrepancy is relatively large, which can improve the performances of densely connected auto-encoder networks without changing the architectures.

---

### Official Review · AnonReviewer3 · 2019-11-13
**Official Blind Review #3**

**Rating:** 3

**Review:**

This paper proposed combined form of flexible activation functions with carefully designed principle of choosing activation functions. It shows some gains on stock price and one standard image task over the baseline.

I'm leaning to reject or give borderline for this paper because (1) the paper don't have comparison with neural architecture search. For example, https://arxiv.org/abs/1710.05941 (Searching for Activation Functions). I don't know what the advantage of this approach compared to searched activation. I guess it's less computation heavy and maybe better motivated. But at least the author should give some pros/cons. (2) the paper has two benchmark, stock price prediction and CIFAR-10. I don't understand why as arch paper, it use such non standard benchmark (stock) and non standard arch (LeNet?). I don't think based these benchmark we can make solid conclusion. (3) These model seems introduce quite a bit more hyper-parameters. It's unclear if it is better than tuning other architecture e.g., batch norm/layer norm/dropout or even just optimizer. For example, "flexible is 0.032" does this parameter generalize to other dataset? Like if the gain is really significant, like resnet over AlexNet, hyperparam doesn't matter. But if it's marginal win over a weak baseline, the how to get the results is important.

Some comments:
In section 2, " back propagation of these activation parameters by stochastic gradient descent can be
done as follows"
Why we need to list the detail backprop formulation here? Are these special? Isn't just autograd?

Can the author explain more for principle 1? What is the "same domain" means here?

**Experience Assessment:**

I have read many papers in this area.

**Review Assessment: Checking Correctness Of Derivations And Theory:**

I assessed the sensibility of the derivations and theory.

**Review Assessment: Checking Correctness Of Experiments:**

I assessed the sensibility of the experiments.

**Review Assessment: Thoroughness In Paper Reading:**

I read the paper at least twice and used my best judgement in assessing the paper.

---

> ### Author Response · Authors · 2019-11-14
> **Response to reviewer #3**
>
> Thank you very much for your feedback.  We would like to address your comments as below:
> (1)	Comparison with neural architecture search: For the paper of searched activation functions (Ramachandran et al. 2017), they actually proposed a function with unbounded domain searched empirically with reinforcement learning, which can be applied to replace ReLu. It is a specific form rather than a general form, and cannot be used to replace sigmoid or tanh, whose output are bounded. Even though they used large datasets and large models with tremendous computing power in Google Brain, their experimental results are not significant with only 5 runs for each setting, while we do at least 50 runs and provide the standard error of the sample means for testing the significances. Therefore, we do not think they proposed a better activation function or method than ours.
> (2)	Non-standard benchmark: We are facing the dilemma of choosing large models and doing more trials for testing the significances. Please be aware that in a lot of scenarios in practice, such as edge computing, we prefer small models. The results for the first experiment in forecasting national stock indices is definitely significant if we are using stacked LSTMs. Further experiments have shown that the performances of models with flexible activation functions surpass that with standard sigmoid for both G7 and G20 countries without randomly choosing the subset.  In the second experiment, we achieved even quite significant improvement over ReLu and significant improvement over PRelu in a large proportion of cases.
> (3)	More hyper-parameters: It is shown in the experiments that even without regularization, the two newly proposed activation functions can achieve significantly better results than Sigmoid and ReLu for muti-variate time series prediction and image compression. Thus, we did not introduced extra hyper-parameters to achieve significant improvements.
> (4)	Backprop formulation: As we proposed a general form of flexible activation function, it is natural to write down the back-propagation formula for this form to show the difference from backprop of weights and bias. In some related papers including the original paper of PRelu, they also provided the backprop formula.  In practice, for functions whose derivatives can not be calculated in Pytorch automatedly, we need to define their backward path manually by using this formula.
> (5)	Explanation of principle 1: For Principle 1 in Section 2.1,  “same domain” means the component activation functions should have the same output domain as the baseline fixed activation functions. For example, when we want to use a flexible combination to replace sigmoid, we need to make sure that the outputs of all the component functions are between 0 and 1.

---

### Author Response · Authors · 2019-11-14
**Response to All Reviewers**

Dear reviewers,

Thanks again for your time and endeavour in reviewing our manuscript. Your feedbacks are very valuable and helpful in improving our work. However, some of your suggestions require large amount of computational power and redesigning of the experiments, and there are too many hyper-parameters to be controlled, which may require much more time for conducting careful experiments and reaching presentable results. We know that the proposed flexible sigmodal activtation function can improve the performance of stacked LSTM models on forecasting G7 and G20 indices as a whole. Also, the regularization effect should be even significant for large models with large amount of flexible activation parameters.

To our knowledge, given a sufficient number of units, a well-designed activation functions can approximate the models with other activation functions to any level of accuracy, which is referred to as universal approximation theorem. Therefore, even a good activation function cannot outperform other ones in any settings of hyper-parameter and model architecture, but some of them are good at handling a particular set of learning tasks with certain model architectures. In addition, training algorithms can also make a difference and have some interactions with the selection of activation functions. We believe we proposed a family of promising flexible activation functions and we will further study them.

---

### Decision · Program_Chairs · 2019-12-19

**Decision:**

Reject

**Comment:**

Main content: Proposes combining flexible activation functions

Discussion:
reviewer 1: main issue is unfamiliar with stock dataset, and CIFAR dataset has a bad baseline.
reviewer 2: main issue is around baselines and writing.
reviewer 3: main issue is paper does not compare with NAS.

Recommendation: All 3 reviewers vote reject. Paper can be improved with stronger baselines and experiments. I recommend Reject.